# Childbirth Satisfaction during the COVID-19 Pandemic in a Hospital in Southwestern Spain

**DOI:** 10.3390/ijerph19159636

**Published:** 2022-08-05

**Authors:** María González-Morcillo, Esther Tiburcio-Palos, Sergio Cordovilla-Guardia, Esperanza Santano-Mogena, Cristina Franco-Antonio

**Affiliations:** 1Servicio Extremeño de Salud, Hospital San Pedro de Alcántara, 10003 Cáceres, Spain; 2Centro José Mª Álvarez, Servicio Extremeño de Salud, 06400 Don Benito, Spain; 3Nursing Department, Nursing and Occupational Therapy College, Universidad de Extremadura, Avda de la Universidad s/n, 10003 Cáceres, Spain; 4Health and Care Research Group (GISyC), Universidad de Extremadura, 10003 Cáceres, Spain

**Keywords:** satisfaction, childbirth, quality, care, COVID-19

## Abstract

Satisfaction, in relation to care received, is a good indicator of quality of care. The objective of this study was to analyze the degree of satisfaction with childbirth and postpartum care as reported by women from one hospital in southwestern Spain during the COVID-19 pandemic. Factors that influenced care were also examined. A cross-sectional study was carried out between the months of October 2020 and February 2021. Satisfaction was measured through the COMFORTS scale, validated in Spanish. A final sample of 116 women was included in the study. The mean age was 32.08 (±4.68) years. A total of 111 (95.69%) women were satisfied or very satisfied with the care received. The median satisfaction score was higher among multiparous women (187 (199–173)) than among primiparous women (174 (193–155.50)) (*p* = 0.003). Differences in satisfaction were found as a function of the use of epidural analgesia, being higher among women who had planned its use but ultimately did not use it (188 (172.50–199.75)) or who planned its use and did (186 (169.50–198)) than among those who had not planned to use epidural analgesia but ultimately received it (173.50 (187.50–146.25)) or those who did not use it, as planned, before childbirth (172 (157–185)) (*p* = 0.020). Overall satisfaction rate between SARS-CoV-2-negative women assisted was high. Parity and use of epidural analgesia were two factors influencing satisfaction scores in our sample.

## 1. Introduction

The opinion of patients regarding their degree of satisfaction is considered a good indicator for the quality of care received, representing 20% of the total value of quality in the European Quality Management Model of the EFQM (European Foundation for Quality Management) [1]. Such opinions are a form of participation by users of health systems, allowing the expression of their perceptions and a valuation of the services [2]. Satisfaction analyses are, therefore, used as instruments to create different health reforms and to improve quality of care. However, the measurement of satisfaction can be challenging because it is a complex and multidimensional concept [3,4].

Satisfaction has been defined in numerous ways over time [5,6]; patient satisfaction can be defined as the positive assessment of a series of complex health actions, based more on the coverage of previous expectations than on feelings themselves [5]. In general, the factors associated with better satisfaction are: (1) continuity of care by the same professional, (2) quality of the professional in terms of the relationship with the patient and adequate communication and (3) qualities of the professional, for example, flexible, informal, interested, friendly and experienced [3,7]. In turn, the provision of personalized attention, based on the needs expressed by patients, leads to high satisfaction with the care received [8].

Childbirth is a very important event in the lives of women, during which numerous physiological and psychological changes occur. It is a unique moment, full of emotions, that each pregnant woman experiences in a different way; therefore, care must be individual, humanized and comprehensive. What occurs during childbirth can mark future behaviors, both positively and negatively [9,10,11]. The satisfaction perceived by pregnant women in relation to the experience of childbirth is influenced by a large number of factors, such as (i) the personal experiences of women, (ii) pain relief, (iii) the support received by family members and health professionals, (iv) the fulfilment of the expectations of women and (v) their participation in decision-making [4,7,12,13,14]; these last two factors are essential. Thus, expectations and differences from the lived experience affect assessments by women regarding the care received during the birth process, as with pain and its management [12].

When analyzing the factors related to the satisfaction of women with their childbirth process, it is important to differentiate between the labor experience (pain, emotional experience, expectations, etc.) and the care received. There are several factors that contribute to feelings of satisfaction or dissatisfaction [7].

According to various studies, between 5 and 7% of women are dissatisfied with the care they received during childbirth [7,12,15,16]. This perception is not specific but can be maintained up to 1 year after birth, influencing future reproductive life and even leading to complaints or legal claims [9,10,12,17]. A positive memory of childbirth increases feelings of achievement, improves self-esteem and is associated with a faster adaptation to motherhood [7]. In contrast, a traumatic childbirth experience has repercussions on the abilities to breastfeed and create an adequate bond. These poor memories can influence future decisions, such as whether the woman prefers a caesarean section in subsequent births or is unable to resume sexual relations [7,18].

The development of the COVID-19 pandemic since 2020 has led to the implementation of new protocols and measures for its control that caused a series of modifications in childbirth care during 2020 and 2021 [19], measures that were adapted to the presence of different incidence peaks. The pandemic itself and the changes in care produced have been associated with a decrease in satisfaction with childbirth care in other countries, not only in SARS-CoV-2-infected women at the time of birth but in general, in all women who gave birth during the pandemic [20]. It is, therefore, important to explore, in our environment, the satisfaction of women with the care they receive when giving birth during this period of change and uncertainty. Therefore, the objective of this study was to determine the degree of satisfaction of pregnant women with the care they received when giving birth and during postpartum hospitalization in a reference hospital in southwestern Spain during the COVID-19 pandemic and the related factors.

## 2. Materials and Methods

This was a descriptive cross-sectional study. The study was approved by the Research Ethics Committee of Badajoz (ref: JBR/jgvp). At all times, the voluntary participation of the participants and their anonymity were ensured, respecting the ethical principles of the Declaration of Helsinki [21]. Prior to data collection, the participants received written information regarding the study and signed an informed consent form.

### 2.1. Setting

The study was carried out at Hospital de Mérida (Badajoz) located in southwestern Spain. This is the health reference hospital for a population of over 164,000 inhabitants, which attends approximately 1400 childbirths per year. The study period was 15 October 2020 to 15 February 2021. The study population included women who gave birth to a live only child within the previous 48 h. The exclusion criteria were as follows: (i) women under 18 years of age, (ii) women with any communication barrier that would make it difficult to understand and complete the questionnaire and (iii) women with a positive real-time polymerase chain reaction (RT-PCR) test result for SARS-CoV-2 in nasopharyngeal exudate at the time of admission. SARS-CoV-2-positive women were not included in the study due to pandemic restrictions limiting access to these women for data collection.

The sample size necessary for the study was estimated on the basis that 90% of the population would be satisfied with the care received, as observed in previous studies [4,7,15]. With a confidence level of 95%, a margin of error of 5% and an estimate of the possible need for replacements of 15%, with an estimated reference population of 375 births during the study period (based on the average number of births at the center), an estimated minimum of 114 women were necessary for this study.

### 2.2. Sample

During the study period, 170 women were invited to participate. Selection was carried out by simple randomization. The random selection process was carried out in the hospitalization unit by means of a randomization sequence that was blinded during the entire sample recruitment period. The random selection was carried out by means of an online computer tool that indicated to the midwives and obstetricians whether or not to offer participation according to the order of admission to the unit. Of the total number of women, 24 refused to participate, so the sample consisted of 146 participants.

### 2.3. Measurement

The following variables were collected:Sociodemographic variables: age, education level (no studies/primary; secondary studies; university/postgraduate studies), stable partner (yes/no) and employment status (student, unemployed, work for others, own-account work);Variables related to care or expectations of childbirth: accompaniment during childbirth and early postpartum(partner/other relative/no), the companion was the same chosen before birth (yes/no), number of professionals who provide care, number of professionals who perform vaginal exams, prenatal education (yes/no), type of feeding desired for the newborn (exclusive breastfeeding, partial breastfeeding, formula feeding), prepartum desire to use epidural analgesia (yes/no), perceived efficacy of the analgesia employed during childbirth (relieved/did not/partial relieved);Satisfaction with the care received during childbirth was measured by the COMFORTS scale [22], translated and validated into Spanish [23]. This scale consists of 40 items grouped into 4 dimensions: (1) Care during the childbirth period. This dimension includes 13 items. All the items included in this area provided information on the care provided to the puerperal woman and her partner or support person during the labour. (2) Postpartum care in the ward. This dimension includes 11 items that explore the care received during the postpartum period, the information and education received by nurses in relation to the care and feeding of the newborn. (3) New-born care. This area includes 10 items and explores woman’s confidence that nurses can meet the needs of her newborn. (4) Logistical aspects and respect for privacy. This dimension includes 6 items and explores satisfaction in relation to the environment (suitability, space, lighting), respect for privacy or quality of the food received; the scale is scored using a Likert scale with a maximum score of 200 points. This scale has good reliability (Cronbach’s alpha = 0.952). This scale categorizes the total degree of satisfaction during the birth process from calculating the total scale score and grouping this within the following ranges: very unsatisfied (40–71 points); unsatisfied (72–103 points); indifferent (104–135 points); satisfied (136–167 points); and very satisfied (168–200 points); andData related to the process of childbirth and the early postpartum period: number of pregnancies, onset of labor (induced/spontaneous), duration of pregnancy (preterm/term/post term), parity (primiparous/multiparous), type of childbirth (eutocic, instrumental, caesarean section), use of epidural analgesia (yes/no), use of nonpharmacological analgesia (yes/no), perineal injury (intact, 1st degree tear, 2nd degree tear, 3rd degree tear, 4th degree tear, episiotomy), time of childbirth (day/night) and feeding of the newborn 48 h after birth (exclusive breastfeeding, partial breastfeeding, formula feeding).

### 2.4. Data Collection

A self-report questionnaire to collect sociodemographic variables and variables related to care or expectations of childbirth and the COMFORTS scale was completed by pregnant women 48 h after giving birth and after signing an informed consent form.

Incomplete or incorrectly answered questionnaires were eliminated from the analyses.

All data related to childbirth and early postpartum processes were collected from the clinical history.

### 2.5. Data Analysis

For the statistical analysis, the distribution of quantitative variables was assessed. For the variables with a nonnormal distribution, the median and the interquartile range [IQR] were used as measures of central tendency and dispersion, respectively. For the variables with a normal distribution, the mean and standard deviation were used as measures of central tendency and dispersion. The satisfaction level was compared through Kruskal–Wallis or Mann–Whitney U test. Spearman’s regression coefficient was used to analyze the association between the duration of the different stages of labor and the level of satisfaction. The scores obtained in the dimensions of the COMFORTS scale were compared through the Friedman test. All analyses were performed using SPSS 24.0 for Windows (SPSS, Chicago, IL, USA), considering *p* values < 0.05 to be significant.

## 3. Results

Of the 146 initial participants, 30 women were eliminated from the analyses because their questionnaires were incomplete, resulting in a final sample of 116 women. No differences were found in the sociodemographic characteristics and variables related to the care received or expectations of care between the women included in the analysis and those excluded for submitting incomplete questionnaires (Appendix A).

The mean age of the participants was 32.08 (±4.68) years, 109 (94.0%) were Spanish, 53 (45.7%) had a university or postgraduate education and all had a stable partner (Table 1).

Further, 52 (44.8%) of the participants were primiparous and for 43 (37.1%), this was their first pregnancy. Forty-six (39.7%) had attended prenatal education classes and 85 (73.3%) had planned to use an epidural as an analgesic measure (Table 2).

Further, 78 (67.2%) of the participants had a eutocic childbirth and 104 (89.7%) were accompanied by their partner during childbirth. Eighty-two (70.7%) women followed their plan regarding the use of epidural analgesia. Only 17 (14.7%) used nonpharmacological pain relief methods. A total of 53 (45.7%) of the women had vaginal examinations performed by three or more different people (Table 2).

Moreover, 91 (78.4%) planned exclusive breastfeeding as the mode of newborn feeding, 11 (9.5%) planned partial breastfeeding to their child and 14 (12.1) planned to feed their newborn with formula.

During early postpartum hospital admission, 113 (97.4%) of the participants were accompanied. Regarding the feeding of the newborn at 48 h after childbirth, 92 (79.3%) were exclusively breastfed, 8 (6.9%) were partially breastfed and 16 (13.8%) were artificially breastfed.

The median overall COMFORTS scale score was 182.50 (164.25–196) points, which fits within the ‘very satisfied’ range of the scale [22]. Dimension 1, “Care during the childbirth period”, was the best valued with a median score of 63 (58–65) points (Figure 1). The median of dimension 2, “Care of the puerperium in the ward”, was 49.50 (44–55) points; dimension 3, “Newborn care”, was 43 (39–50) points; and 28 (24.25–30) points for dimension 4, “Logistical aspects and respect for privacy”. When categorizing the global satisfaction as proposed by the scale, 85 (75.3%) felt very satisfied with the care received throughout the process, 26 (22.4%) felt satisfied, 4 (3.5%) felt indifferent with the care received and 1 (0.9%) felt dissatisfied.

When analyzing the relationship of the baseline variables and childbirth process with the COMFORTS scale score, greater satisfaction was found among multiparous women than among those for whom this was their first birth (187 (199–173) vs. 174 (193–155.50), *p* = 0.003) (Table 3). Additionally, those who planned to use an epidural, whether or not they actually used it during childbirth, had higher COMFORTS scale scores (186 (169.50–198) and 188 (172.50–199.75), respectively) than did those who planned to give birth without an epidural, whether they followed this plan (172 (157–185)) or did not (173.50 (187.50–146.25)) (*p* = 0.020) (Table 3).

The duration of the second and third stages of labor did not correlate with degree of satisfaction (*p* = 0.123 and *p* = 0.326, respectively).

## 4. Discussion

The arrival of COVID-19 at the beginning of 2020 meant a change in the global health-care model. This change also led to strong changes in the care of women during pregnancy, childbirth and the postnatal weeks, and basic elements of the midwife–woman relationship, such as meeting in person and providing a comforting touch, were upended in an attempt to maintain distance and reduce cross-infection [24]. In Spain, during the first months of the pandemic, the care protocols for women underwent several changes, which meant a high change in the measures to be adopted in maternity services, both for SARS-CoV-2-positive women and for others [24,25] These changes during these first months meant an increase in unnecessary interventions and severe limited visiting, even certain limitations to the labor or postpartum companionship [24].

This study was carried out when in our territory, the second wave gave way to a third wave. At the end of 2020 and the beginning of 2021, restrictions returned, like those at the beginning of the pandemic [26]. The incidence in the reference territory of the Mérida Hospital was on the rise and, during the development of this study, our territory reached a high incidence [26]. This meant maintaining and reinforcing restrictions within the hospital, not only for COVID-19-infected women, but preventively for all others, since vaccination had not yet been started in pregnant women [26]. These restrictions and changes in care are associated with low satisfaction with care received [20,27].

The results of our study show that, in total, 75.3% of the women in our sample were very satisfied with the care received, with only 0.9% of women reporting that they were dissatisfied with the care and 3.5% reporting that they were indifferent. Therefore, it appears that the rate of satisfaction of SARS-CoV-2-negative women with the care received in this hospital was high; high rates of satisfaction can also be found in other hospitals in Spain [4,7,12,15]. On the other hand, one study found lower satisfaction in women cared for during the pandemic compared to those who gave birth before the pandemic, being lower in SARS-CoV-2-positive women [20]. Nevertheless, a study carried out in our country did not find differences in satisfaction between women who gave birth before or during the pandemic [28]. In our case, we cannot determine if the care changes that occurred in our center during the pandemic had an impact on satisfaction because there are no previous data on the satisfaction of women with the care they received at our center before; however, the satisfaction of SARS-CoV-2-negative women with the care received was high during the study period, which occurred during the COVID-19 pandemic.

The analysis of the four dimensions of the COMFORTS scale showed that dimension 1, related to childbirth care, was the highest valued, while dimension 3, related to newborn care, was the lowest valued. Another study carried out in our country also found that this dimension was the lowest valued [29]. This analysis of the dimensions will allow us to focus our efforts to implement measures that allow us to improve the quality of care.

When analyzing the factors related to satisfaction, satisfaction was higher among women with a previous birth experience. The same is observed in other studies, i.e., multiparity is associated with greater satisfaction [13,15,30]; as a potential explanation, multiparous women may have more realistic expectations of the birthing process, with the differences between expectations and reality being lower.

Women who had planned to use an epidural reported greater satisfaction, whether or not they had used an epidural during the birthing process. The evidence regarding epidural use and satisfaction is unclear; while a study found that the use of epidural analgesia was associated with greater satisfaction [14], in other studies, women who do not use epidural analgesia were more satisfied [4,31]. In this study, however, the use itself was not associated with satisfaction but rather having planned to use it. It is possible that women who plan to use an epidural have more realistic expectations of pain during childbirth. Interestingly, of all the groups, the women who had planned to use an epidural but did not do so reported the highest levels of satisfaction, leading us to think that this result could be related to the perception of pain during childbirth as the key element of satisfaction.

Finally, in this study, the accompaniment of women during postpartum hospitalization was associated with greater or lesser satisfaction, finding that those who were accompanied by a person other than their partner were the least satisfied, even less satisfied than those who were not accompanied during this period. Other previous studies [14,32] found that being accompanied during childbirth or the postpartum period by family and friends is associated with higher satisfaction; however, in this study, being accompanied by someone other than the partner reduced satisfaction. We cannot affirm the cause of this result, although it could be because the actual companion was not who was chosen prior to childbirth; we found somewhat lower satisfaction scores among women whose companion was not who they had initially chosen. A prospective longitudinal study carried out at the beginning of the pandemic found that restrictions on companions negatively affected satisfaction, along with other factors, such as the stress perceived by the woman or the use of a mask [27]. No other factors related to satisfaction were found in this study; however, other previous studies found that a lower level of education is associated with greater satisfaction [14], that the type of childbirth influences satisfaction [30,32] and that spontaneous onset of labor is associated with higher satisfaction [14,30]. In other studies, attendance at prenatal education classes was associated with greater satisfaction [16]; in our study, even though the COMFORTS scale scores of women who had attended prenatal education were higher than those of women who did not attend such classes, the differences were not significant.

## 5. Limitations

This study has considerable limitations, specific to the study design chosen. The main objective of the study was to know the degree of satisfaction and, therefore, the study was cross-sectional. This design limits the establishment of causal relationships in the associations found.

During the period of this study, the strong restrictions and care protocols limited access to SARS-CoV-2-infected women and this fact influenced the inclusion criteria that limited the possibility of determining the satisfaction of women who tested positive for SARS-CoV-2, women who experienced substantial limitations that could have impacted their care and, ultimately, their satisfaction.

On the other hand, it is evident that this study cannot be extrapolated to other populations or centers because the care measures, restriction or protocols may differ. In addition, the associations found, although some coincide with the previous literature, do not establish causal relationships due to the chosen design.

In total, 20.5% of the questionnaires were partially or incorrectly completed; this percentage exceeded the expected replacement estimate, which could have influenced the validity of the results. This high incompletion rate suggests that the validated scale used to measure satisfaction may not be the most suitable for the study population.

Nevertheless, this type of study allows us to establish the points of improvement in care and those factors that we could influence. It is necessary to carry out future studies that analyze the impact of changes made and satisfaction in COVID-19-positive women.

## 6. Conclusions

The satisfaction rate of SARS-CoV-2-negative women who gave birth at Hospital de Mérida during the study period was high and a very low percentage of women were dissatisfied with the care received. The care received during labor was the highest valued and newborn care the lowest.

Having previous childbirth experience and having chosen an epidural as analgesia before childbirth were associated with higher satisfaction. In contrast, being accompanied during the early postpartum period by a companion other than a partner was associated with lower COMFORTS scale scores. Factors, such as the number of professionals who provided care, attendance at prenatal education classes or the type of childbirth, did not influence the assessment of satisfaction.

## Figures and Tables

**Figure 1 ijerph-19-09636-f001:**
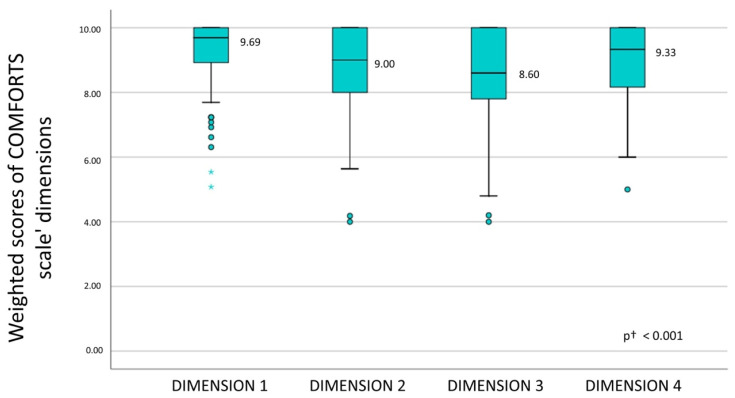
Scores of COMFORTS scale’ dimensions. † Friedman test.

**Table 1 ijerph-19-09636-t001:** Sociodemographic characteristics of the participants.

Variables	N = 116
**Age** $\bar{x}$ **(SD)**	32.08 (4.68)
**Country of origin n (%)**	
Spain	109 (94.0)
Other	7 (6.0)
**Education n (%)**	
No studies/primary	40 (34.5)
Secondary	53 (45.7)
University/postgraduate	23 (19.8)
**Employment situation n (%)**	
Student	2 (1.7)
Unemployed	41 (35.3)
Work for others	65 (56.0)
Own-account work	8 (6.9)
**Stable partner n (%)**	116 (100.0)

$\bar{x}$: mean; SD: standar deviation.

**Table 2 ijerph-19-09636-t002:** Variables related to the childbirth.

Variables	N = 116
**Gravidity n (%)**	
Primigravida	43 (37.1)
Multigravida	73 (62.9)
**Parity (n%)**	
Primiparous	52 (44.8)
Multiparous	64 (55.2)
**Planned epidural analgesia before childbirth**	85 (73.3)
**Maternal Education n (%)**	
Yes	46 (39.7)
No	70 (60.3)
**Onset of labor n (%)**	
Induced	44 (37.9)
Spontaneous	72 (62.1)
**Duration of pregnancy, n (%)**	
Preterm	5 (4.3)
Term	67 (57. 8)
Post term	44 (37.9)
**Duration of dilation, (minutes) Median (IQR)**	170 (67.50–266.25)
**Duration of expulsive efforts, (minutes) Median (IQR)**	52.50 (20–110)
**Type of childbirth, n (%)**	
Eutocic	78 (67.2)
Instrumental	14 (12.1)
Caesarean section	24 (20.7)
**Number of professionals who provided care, n (%)**	
1–2	37 (31.9)
3–5	67 (57.8)
>5	26 (10.4)
**Number of professionals who performed vaginal examination, n (%)**	
0–2	63 (54.3)
3–5	52 (44.8)
>5	1 (0.9)
**Perineal injury, n (%)**	
No perineal trauma	41 (35.3)
First degree tear	24 (20.7)
Second degree tear	33 (28.4)
Episiotomy	18 (15.5)
**Time of birth, n (%)**	
Day	50 (43.1)
Night	66 (56.9)
**Labor Companion, n (%)**	
Partner	103 (88.8)
Other relative	4 (3.4)
Nobody	9 (7.8)
**The companion was chosen prior to childbirth, n (%)**	104 (89.7)
**Use of epidural analgesia during childbirth, n (%)**	87 (75)
**Epidural effectiveness (n = 87), n (%)**	
Relieved	61 (70.1)
Did not relieve	2 (2.3)
Partial relief, use of bolus	24 (27.6)
**Epidural used as planned, n (%)**	
Used as planned	69 (59.5)
Did not use as planned	13 (11.2)
Used without initial plan	18 (15.5)
Did not use, planned do it	16 (13.8)
**Use of nonpharmacological pain relief methods, n (%)**	17 (14.7)

IQR: Interquartile range.

**Table 3 ijerph-19-09636-t003:** Analysis of factors associated with COMFORTS scale scores.

Variable	Median (IQR)	*p*
**Education level**		
No education/primary (n = 40)	184 (174–198)	0.163 ^1^
Secondary education (n = 53)	193 (157.5–199)	
Higher education (n = 23)	179 (162–187)	
**Gravidity**		
Primigravida (n = 43)	179 (195–157)	0.158 ^2^
Multigravida (n = 73)	185 (199–167)	
**Parity**		
Primiparous (n = 52)	174 (193–155.5)	0.003 ^2^
Multiparous (n = 64)	187 (199–173)	
**Maternal education**		
Yes (n = 46)	185 (196–158.5)	0.861 ^2^
No (n = 70)	180.5 (199–166.8)	
**Onset of labor**		
Induced (n = 44)	179.5 (160–196.8)	0.648 ^2^
Spontaneous (n = 72)	183.5 (165.8–196)	
**Duration of pregnancy**		
Preterm (n = 5)	187 (164–198)	0.556 ^1^
Term (n = 67)	179 (172–190)	
Post term (n = 44)	179.5 (160.5–195.5)	
**Type of chilbirth**		
Eutocic (n = 78)	182.5 (168.8–196.5)	0.090 ^1^
Instrumental (n = 14)	191.5 (176.8–200)	
Caesarian section (n = 24)	171.5 (157–193.5)	
**Number of professionals providing care**		
1–2 (n = 37)	182 (169.5–199)	0.703 ^1^
3–5 (n = 67)	180 (159–196)	
>5 (n = 26)	184.50 (174–196)	
**Number of professionals who performed a vaginal examination**		
0–2 (n = 63)	182 (162–199)	0.338 ^1^
3–5 (n = 52)	181.5 (167.3–196)	
>5 (n = 1)	200 (200–200)	
**Perineal injury**		
No perineal trauma (n = 41)	179 (160–196)	0.337 ^1^
First degree tear (n = 24)	191 (179.5–199.8)	
Second degree tear (n = 33)	183 (160.5–198)	
Episiotomy (n = 18)	176.5 (151–190)	
**Labor companion**		
Partner (n = 103)	183 (162–196)	0.269 ^1^
Other relative (n = 4)	172.5 (168.5–178)	
Nobody (n = 9)	196 (168.5–199.5)	
**The companion was chosen prior to childbirth**		
Yes (n = 104)	182.50 (162.5–196)	0.293 ^2^
No (n = 12)	170 (168–170)	
**Time of birth**		
Day (n = 50)	185.50 (165.8–198.3)	0.254 ^2^
Night (n = 66)	179.50 (161.5–196)	
**Use of epidural analgesia**		
Yes (n = 87)	183 (164–196)	0.830 ^2^
No (n = 29)	182 (165–197)	
**Epidural effectiveness**		
Relieved (n = 61)	183 (163–196)	0.619 ^1^
Did not relieve (n = 2)	169 (153–169)	
Partial relief, use of bolus (n = 24)	179.5 (166.5–199.8)	
**Epidural use as planned**		
Used as planned (n = 69)	186 (169.5–198)	0.020 ^1^
Did not use as planned (n = 13)	172 (157–185)	
Used without initial plan (n = 18)	173.5 (187.50–146.3)	
Did not use, planned do it (n = 16)	188 (172.5–199.8)	
**Use of nonpharmacological analgesic methods**		
Yes (n = 17)	179 (162–195.5)	0.769 ^2^
No (n = 99)	183 (165–196)	
**Companion on the hospital ward during the postpartum period**		
Partner (n = 111)	182 (162–196)	0.035 ^1^
Other relative (n = 2)	177 (175–177)	
Nobody (n = 3)	200 (200–200)	

^1^ Kruskal–Wallis test; ^2^ Mann–Whitney U test; IQR: Interquartile range.

## Data Availability

The datasets from the current study are available from the corresponding author on reasonable request.

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
