# Peer review of "Childbirth Satisfaction during the COVID-19 Pandemic in a Hospital in Southwestern Spain"

_ijerph, 2022, doi:10.3390/ijerph19159636_

Round 1
Reviewer 1 Report
The study is presented as a cross-sectional study title “Childbirth satisfaction during the covid-19 pandemic in a hospital in southwestern Spain”
The aim of this study is to determine the degree of satisfaction of pregnant women with the care they received when giving birth and during postpartum hospitalization in a reference hospital in southwestern Spain during the COVID-19 pandemic.
The study has been carried out according to standards and has been described very well. As the topic is very interesting, the manuscript presents several limitations that in the current state prevent its publication.
Abstract
Point 1: The aim does not match what was stated in the introduction section (lines 79-81). There isn´t conclusion (line 26)
Methods
Point 2: An explanatory phrase must be added about the type of Hospital, number of births / years, area of influence, or information of interest to be able to estimate the magnitude of the center. Lines 89-90.
Point 3: Exclusion criteria, is better to specify in this form “with a positive real-time polymerase chain reaction (RT-PCR) test result for SARS-CoV-2 in nasopharyngeal exudate at the time of admission”. If not, please justify it. Line 93
Point 4: Sample size must be justified. The reference population of 375 births are in the study period of other years? Please clarify in order to calculate the sample size. Lines 99-100,
Point 5: The authors must justify why if the estimated sample was 114, why they reached 170, if only the estimated % of losses was 15%. Lines 103-104.
Point 6: The variables need to be explained to understand the results´ tables, for example: I do not understand third-party account. Partner status? Accompaniment? (where in labour, at admission, etc…) Please explain in a few words something information about how the variables were included.
Point 7: As a supplementary file include the COMFORTS scale in Spanish. Lines 115.
Point 8: Authors should explain the categories of variables, for example, parity (primiparous/multiparous), etc. Perinatal injury must be explained, and which scaled has been used to did it. Lines 120-123.
Results
Point 9: In Table 1, epidural and feeding mode aren´t sociodemographic variables, please they must be put in another table. Please, change new-born to newborn.
Pont 10: Please change nationality to country of origin. Include 1 decimal number to coherence. (Nationality results and employment situation-active, third-party account)
Point 11: There are variables that are not included in the measurement section (onset of labour, end of pregnancy, duration…).
Point 12: The authors indicate that 43 women were primiparous and that 52 had a first birth. It would be better to describe the number of pregnancies and parity separately, because this is incongruous. All this paragraph is not reflected in Table 1 as indicated. Lines 144-146.
Point 13: In table 2, what´s complete in perineal injury? A D1, D2? Please clarify it.
Point 14: I recommend using labour companion, and labour and not delivery.
Point 15: The format of the tables is not in line with the journal rules. Please verify it.
Point 16: The title of the table 3 must be include the scale used. Again, primiparous and first delivery are incongruous.
Point 17: In table 3, was the companion…. Is it an answer? Is it an affirmative phrase? Please verify it.
Point 18: Figure 1 needs further explanation. If it has 4 dimensions, why only one is represented? Which of all is represented? It would be better to represent the mean of the 4 dimensions with their confidence intervals.
Discussion
Point 19: In my point of view, the results should not be reflected in the discussion so specifically. The first paragraph should justify the study in its context and that it contributes to the existing evidence. Lines 178-183.
Point 20: In the second paragraph, it would be interesting to place the reader, to describe the moment in which the study was being carried out to know the protocols taken in the birth, wave number, if there was vaccination in pregnant women, etc., data of interest that provide a context at the time of the pandemic. This bibliography may be useful:
1-Coxon, K.; Turienzo, C.F.; Kweekel, L.; Goodarzi, B.; Brigante, L.; Simon, A.; Lanau, M.M. The impact of the coronavirus (COVID-19) pandemic on maternity care in Europe. Midwifery 2020, 88, 102779.
2-Berghella, V.; Brenna, L. COVID-19: Labor, Birth, and Postpartum Issues and Care—UpToDate. Available online: https:
//www.uptodate.com/contents/covid-19-labor-birth-and-postpartum-issues-and-care
3-Int. J. Environ. Res. Public Health 2022, 19, 5482. https://doi.org/10.3390/ijerph19095482
Point 21: In line 200, the authors refer other studies but only reported reference 14.
Conclusion
Point 22: The conclusion should reply in a general way to the objective set, the study time should not be used, and it´s possible summarize it in two sentences.
Author Response
Thank you very much for the comments and for all the suggestions made, which we are sure improves the quality of the manuscript. We detail the changes made based on your suggestions
The study is presented as a cross-sectional study title “Childbirth satisfaction during the covid-19 pandemic in a hospital in southwestern Spain”
The aim of this study is to determine the degree of satisfaction of pregnant women with the care they received when giving birth and during postpartum hospitalization in a reference hospital in southwestern Spain during the COVID-19 pandemic.
The study has been carried out according to standards and has been described very well. As the topic is very interesting, the manuscript presents several limitations that in the current state prevent its publication.
Abstract
Point 1: The aim does not match what was stated in the introduction section (lines 79-81). There isn´t conclusion (line 26)
Changes have been made in the abstract and introduction section to match the study objective (15-16 and 87). The conclusion has been added to the abstract (line 29)
Methods
Point 2: An explanatory phrase must be added about the type of Hospital, number of births / years, area of influence, or information of interest to be able to estimate the magnitude of the center. Lines 89-90.
The setting section has been modified (Line 95 -98)
Point 3: Exclusion criteria, is better to specify in this form “with a positive real-time polymerase chain reaction (RT-PCR) test result for SARS-CoV-2 in nasopharyngeal exudate at the time of admission”. If not, please justify it. Line 93
We have been followed your recommendation
Point 4: Sample size must be justified. The reference population of 375 births are in the study Period of other years? Please clarify in order to calculate the sample size. Lines 99-100,
This aspect has been clarified in the text (line 110)
Point 5: The authors must justify why if the estimated sample was 114, why they reached 170, if only the estimated % of losses was 15%. Lines 103-104.
The estimated size indicated the minimum number but the researchers decided to request the participation of as many women as possible during the study period
Point 6: The variables need to be explained to understand the results´ tables, for example: I do not understand third-party account. Partner status? Accompaniment? (where in labour, at admission, etc…) Please explain in a few words something information about how the variables were included.
The measurement section has been modified (line 118 – 148)
Point 7: As a supplementary file include the COMFORTS scale in Spanish. Lines 115.
We do not have the intellectual property of the questionnaire for it, but in the reference of the validation study you can see it in full
Point 8: Authors should explain the categories of variables, for example, parity (primiparous/multiparous), etc. Perinatal injury must be explained, and which scaled has been used to did it. Lines 120-123.
The measurement section has been modified (line 118 – 148)
Results
Point 9: In Table 1, epidural and feeding mode aren´t sociodemographic variables, please they must be put in another table. Please, change new-born to newborn.
Table 1 has been changed
Pont 10: Please change nationality to country of origin. Include 1 decimal number to coherence. (Nationality results and employment situation-active, third-party account)
Your suggestions have been followed in the text
Point 11: There are variables that are not included in the measurement section (onset of labour, end of pregnancy, duration…).
Point 12: The authors indicate that 43 women were primiparous and that 52 had a first birth. It would be better to describe the number of pregnancies and parity separately, because this is incongruous. All this paragraph is not reflected in Table 1 as indicated. Lines 144-146.
Thank you for showing us the error, the results section has been reviewed and this data has been corrected (line 181-183)
Point 13: In table 2, what´s complete in perineal injury? A D1, D2? Please clarify it.
Table 2 has been modified
Point 14: I recommend using labour companion, and labour and not delivery.
Your suggestions have been followed in the text
Point 15: The format of the tables is not in line with the journal rules. Please verify it.
The tables has been formatted
Point 16: The title of the table 3 must be include the scale used. Again, primiparous and first delivery are incongruous.
Table 3 has been changed
Point 17: In table 3, was the companion…. Is it an answer? Is it an affirmative phrase? Please verify it.
This has been verified
Point 18: Figure 1 needs further explanation. If it has 4 dimensions, why only one is represented? Which of all is represented? It would be better to represent the mean of the 4 dimensions with their confidence intervals.
The figure 1 represented the total satisfaction. We have changed this figure to show the scores of the 4 dimensions and compare which one was better valued
Discussion
Point 19: In my point of view, the results should not be reflected in the discussion so specifically. The first paragraph should justify the study in its context and that it contributes to the existing evidence. Lines 178-183.
Point 20: In the second paragraph, it would be interesting to place the reader, to describe the moment in which the study was being carried out to know the protocols taken in the birth, wave number, if there was vaccination in pregnant women, etc., data of interest that provide a context at the time of the pandemic. This bibliography may be useful:
1-Coxon, K.; Turienzo, C.F.; Kweekel, L.; Goodarzi, B.; Brigante, L.; Simon, A.; Lanau, M.M. The impact of the coronavirus (COVID-19) pandemic on maternity care in Europe. Midwifery 2020, 88, 102779.
2-Berghella, V.; Brenna, L. COVID-19: Labor, Birth, and Postpartum Issues and Care—UpToDate. Available online: https: //www.uptodate.com/contents/covid-19-labor-birth-and-postpartum-issues-and-care
3-Int. J. Environ. Res. Public Health 2022, 19, 5482. https://doi.org/10.3390/ijerph19095482
Point 21: In line 200, the authors refer other studies but only reported reference 14.
Thank you for your suggestions and resources contributed to improve the discussion. The discussion section has been modified
Conclusion
Point 22: The conclusion should reply in a general way to the objective set, the study time should not be used, and it´s possible summarize it in two sentences.
The conclusion section has been modified
Reviewer 2 Report
Dear Authors
Thank you for the opportunity to review this work.
In my opinion, there are some limits that are not well addressed in the manuscript.
There is a need to state the results more clearly. I believe that a more in-depth data analysis is also needed.
On the other hand, not taking into account COVID positive women I think is a mistake that affects your results and therefore cannot affirm what you say in the discussion.
In the attached file I propose further changes.
I hope it will help you to improve your work.
Best regards and good luck.
Author Response
Thank you for the opportunity to review this work.
In my opinion, there are some limits that are not well addressed in the manuscript.
There is a need to state the results more clearly. I believe that a more in-depth data analysis is also needed.
On the other hand, not taking into account COVID positive women I think is a mistake that affects your results and therefore cannot affirm what you say in the discussion.
In the attached file I propose further changes.
I hope it will help you to improve your work.
Best regards and good luck.
Thank you for your review of the manuscript and for your suggestions.
Due to the design of the study, we do not consider it appropriate to carry out another type of analysis. We are aware of the limitations of the study and the associations found and the limitations section has been updated.
We also think that knowing the opinion of covid-19 positive women was important. The limitations and restrictions of the time of the study made it difficult for the researchers to access these users. This information has been added to the limitations section.
In relation with changes proposed in the text:
Abstract
The changed proposed has been followed.
Methods
Line 91. More information about the moment of the studio has been added along the manuscript, methods and discussion address this information
Line 93. This has been commented above
Line 121. Measurement section has been modified
Results
The results section has been modified following your recommendations
Discussion
Discussion section has been modified
Limitations.
This section has been also changed
Reviewer 3 Report
Estimates Editor:
Thank you for allowing me to review this manuscript. The manuscript is methodologically correct, although the design is simple and its relevance is limited. I know the subject because I have carried out a similar study and we are currently carrying out a multicentric study on this topic. They use a scale little used in our country, although it is validated. Perhaps this is the most innovative aspect. Best regards
Author Response
Thank you for your comments on the manuscript.
Round 2
Reviewer 2 Report
Thank you again for taking my input into consideration.
I attach again new recommendations.
It seems that the authors are determined to show that the women they have treated in their hospital have high satisfaction. It should be noted that such claims cannot be made.
The limitations are stated, although it is not properly justified why COVID-19 positive women were not included in the study.
Good luck with your work

Author Response
Dear reiewer,
Thank your for your new revision.
Regarding the fact of not including positive COVID-19 women, as we have already explained, the strong restrictions at the time made it impossible for researchers to access.
Please see the attachment to see the replies to all your comments.
We hope you find them suitable.
Best regards
